# Parents' work–family conflict and parent–child relationship: The mediating role of parenting burnout and the moderating role of self-compassion

Jing Liang[¤a*☯], Zongping Chen[☯]

Chongqing Normal University, Chongqing, China

☯ These authors contributed equally to this work.
¤a Current address: Southwest University, Chongqing 401331, China.
* 1330968532@qq.com

## Abstract

In today's fast-paced society, balancing work and family has become a key challenge affecting individual well-being, particularly for working parents. The conflict between these roles not only impacts personal mental health but also strains family dynamics, especially the parent–child relationship. The main objective of this study is to explore the impact of work–family conflict on the parent–child relationship, with a focus on the mediating role of parenting burnout and the moderating role of self-compassion. The findings of a sample of 818 working parents revealed that work–family conflict negatively affects parent–child relationships and increases parenting burnout, thus further damaging these relationships. However, self-compassion significantly mitigates these negative effects, reducing the risk of parenting burnout. This study highlights the importance of fostering self-compassion as a potential intervention to protect the parent–child relationship in the context of the work–family relationship.

## Introduction

For adults in modern society, work and family are important areas of life, and their conflicts affect an individual's mental health [1]. Work-family conflict (WFC) refers to the difficulty faced by an individual in attempting to reconcile and balance the different needs associated with the work and family spheres in terms of emotions and behaviors, which results in role conflict [2]. WFC has been extensively studied, with numerous investigations addressing its antecedents [3] and its effects on general parental well-being [4]. Additionally, the consequences of work-family conflict are also expected to exert an impact on the quality of the parent-child relationship. For example, recent studies have highlighted the link between work–family conflict and the parent–child relationship [5], although the mechanism of its intrinsic influence remains unclear.

Previous studies have demonstrated that work–family conflict affects parent–child relationships. One possible explanation is that work–family conflict depletes personal resources, thereby reducing the ability to focus on the emotions and needs of others [6]. In the realm

**Data availability statement:** In accordance with PLOS ONE's data policy, we have uploaded the datasets used in this study to the figshare public database, which can be accessed at https://figshare.com, DOI: 10.6084/m9.figshare.28190315. This ensures that the data underlying our findings are freely available to other researchers.

**Funding:** The 2024 Chongqing Humanities and Social Sciences Research Program (24SKSZ021) and the 2021 Chongqing Education Reform Program (213136)

**Competing interests:** The authors declare that the research was conducted in the absence of any commercial or financial relationships that could be construed as a potential conflict of interest.

of parent-child relationships, this effect implies that work–family conflict may result in the dedication of less time to parent–child activities [7]. Based on a survey of parents with children aged 4–18, Roeters and Van Houdt found that the time parents spend with their children mediates the relationship between work demands and the quality of the parent–child relationship [8]. Thus, work–family conflict affects the quality of the parent–child relationship. However, most previous studies focusing on the impact of work–family conflict on the parent-child relationship [8,9] have failed to explore the factors influencing this relationship. Parenting burnout could play a key role in shaping these associations.

Parenting burnout refers to experiences of emotional dysregulation regarding parenting and encompasses a three-dimensional syndrome of emotional exhaustion, emotional detachment, and a low sense of personal fulfillment; such burnout can be viewed as an emotional response within the parenting subsystem that occurs when parents experience chronic parenting stress [10,11]. From a theoretical perspective, work–family conflict, due to its resource-draining nature, is likely a significant factor contributing to parenting burnout [10,11]. Evidence from previous cross-sectional studies has confirmed a positive association between work–family conflict and parenting burnout symptoms [12,13]. Studies have shown that parenting burnout is related to poorer-quality parent-child relationships [14,15]. Nevertheless, to the best of our knowledge, the indirect effect of work–family conflict on parent–child relationships through parenting burnout has yet to be explored.

As previously mentioned, work–family conflict can negatively impact the family domain, largely due to its consumption of a significant portion of an individual's limited resources [16]. According to conservation of resources (COR) theory, people must rely on limited resources to cope with stressful situations, and the resource depletion caused by work–family conflict makes it difficult for individuals to cope, resulting in adverse effects [17]. However, the theory also emphasizes that an influx of additional resources could moderate the relationship. Therefore, it becomes particularly important to explore additional resources to buffer the adverse effects of work–family conflict. For example, research has reported that social support can buffer the adverse effects of WFC [18]. However, to our knowledge, much of the prior research has explored resources external to the individual, such as social support, whereas the internal resources of individuals must also be considered [19].

With the rise of positive psychology, self-compassion has attracted the attention of researchers as a positive intraindividual psychological resource. Studies on self-compassion have emphasized the aspects of self-concern and self-empathy [20,21]. But the study of self-compassion in relation to work–family conflict is still in its early stages. As an emotion-based resource, self-compassion helps individuals manage stress related to coping and performance [22]. Those with higher levels of self-compassion are more effective in handling the daily challenges and frustrations of life by practicing self-kindness, maintaining positivity, and recognizing shared humanity. As a result, they experience less resource depletion and reduced work–family conflict [23]. Consequently, individuals with greater self-compassion are less likely to have their parent–child relationship impacted by work–family conflict and are less prone to parenting burnout in the face of similar work–family conflict situations.

## Materials and methods

### Ethical statement

All participants in this study voluntarily participated, and only responses related to age, gender, and relevant questionnaires were collected. No personal information, such as names or contact details, was gathered. This study received approval from the Ethics Committee of Chongqing Normal University (Approval Number: CNU-EDU-20240508-003).

## Participants

The focus of this study is on working parents; thus, we distributed questionnaires mainly in this community via online links. A total of 857 working parents completed the questionnaires during the data collection period, which lasted for approximately one month (June–July 2024). The data of 39 participants were excluded because of short response times or regular responses. The data provided by a total of 818 participants, aged 25–55, were thus included in subsequent analyses, for a response rate of 95.44%. The study was conducted in line with the local research ethics committee of Chongqing Normal University, and all participants signed informed consent forms. The participants were fully informed about the purpose and procedures of the study. The consent form clearly indicated their right to withdraw from the study at any time without facing any negative consequences. Participants' privacy and confidentiality were protected. The researchers assured the participants that their personal information would remain confidential and that the data would be anonymized to the greatest extent possible.

## Procedure

A questionnaire survey was used to collect information concerning demographic variables, WFC, parenting burnout, self-compassion, and the parent–child relationship. The participants were informed of the requirements of the study under the guidance that they were provided. They were also informed that the investigation was anonymous and confidential and that they had the right to withdraw from this research at any time without explanation. The questionnaire was distributed online via the popular Chinese professional survey website Wenjuanxing (www.wjx.cn).

## Instruments

**Work–family conflict.**  To evaluate work–family conflict, the Chinese version of the Work–Family Conflict Scale (WAFCS) developed by Ge was utilized [24,25]. This scale consists of 10 items across two dimensions: "work–family conflict" (WFC) and "family–work conflict" (FWC). This 10-item scale includes statements such as "I don't spend enough time with my family because of my job." Participants indicated their levels of agreement with these items on a 7-point scale ranging from 1 (strongly disagree) to 7 (strongly agree). Higher scores signify higher levels of WFC. The scale exhibited excellent internal consistency; the Cronbach's α coefficients were 0.90 for the total scale, 0.87 for the work-to-family conflict subscale and 0.88 for the family-to-work conflict subscale. In this study, data pertaining to the work-to-family conflict dimension were included for further analysis.

**Parent–child relationship.**  The parent–child relationship was measured via the Chinese version of Pianta's (1992) Child–Parent Relationship Scale (CPRS) [26,27]. This scale consists of 26 questions across three dimensions: closeness (10 items), conflict (12 items), and dependence (4 items). However, owing to the low reliability exhibited by the dependence dimension, as noted by Pianta[26], only the closeness and conflict dimensions were considered in this research. Closeness measures a parent's feelings of affection toward and open communication with his or her child (e.g., "I share an affectionate, warm relationship with my child"); in contrast, the conflict dimension assesses a parent's perception of negativity and discord in his or her relationship with his or her child (e.g., "My child sees me as a source of punishment and criticism"). The participants rated each item on a 5-point scale ranging from 1 (definitely does not apply) to 5 (definitely applies), in which context higher scores indicated higher levels of closeness or conflict. The Cronbach's α coefficient for this scale was 0.87.

**Parenting burnout.** Parental burnout was assessed via the Parental Burnout Assessment (PBA) [28], which was validated for use in China by Cheng et al. [29]. Parents rated the frequency of their experiences of parental burnout, including their feelings of exhaustion and irritability related to parenting, on a 7-point Likert scale (ranging from 0 = never to 6 = daily) based on 21 items such as "I feel completely run down by my role as a parent" and "I cannot take being a parent anymore." Total scores on this scale ranged from 0 to 126, with higher scores indicating more severe parental burnout. The Cronbach's α coefficient for this scale was 0.96.

**Self-compassion.** The Chinese version of the Self-Compassion Scale (SCS) was used to measure an individual's overall level of self-compassion [21,30]. This 26-item scale includes three positively framed subscales—self-kindness, common humanity, and mindfulness—which can be combined to form a positive self-compassion score (e.g., "I'm kind to myself when I'm experiencing suffering"). Additionally, three negatively framed subscales—self-judgment, isolation, and overidentification—can be combined to form a negative self-compassion score (e.g., "I'm disapproving and judgmental about my own flaws and inadequacies"). The participants rated each item on a 5-point Likert scale ranging from 1 (almost never) to 5 (almost always). The overall score was calculated as the average of the six subscale scores, in which context higher scores indicated greater self-compassion. The Cronbach's α coefficient for this scale was 0.86.

## Statistical analysis

In the present study, all analyses were performed with the assistance of SPSS 26.0 software. Initially, descriptive statistics and correlation analyses were conducted to illustrate the distributions of and interrelationships among the variables. The moderated mediation model was subsequently tested with the assistance of Hayes's PROCESS macro, which is based on 5000 random samples, to determine bootstrap confidence intervals (CIs) [31]. Significant effects were identified on the basis of 95% confidence intervals excluding zero [32]. The number of children was included as a control variable. All variables were standardized prior to the analysis.

## Aim of the study

This study aimed primarily to explore the mediating role of parenting burnout in the link between work–family conflict and the parent–child relationship, as well as the moderating effect of self-compassion on this connection. Specifically, the study proposed and tested the following hypotheses: (a) work–family conflict significantly and negatively predicts the parent–child relationship; (b) parenting burnout acts as a mediator in the relationship between work–family conflict and the parent–child relationship; and (c) self-compassion moderates the association between work–family conflict and the parent–child relationship. The proposed research model illustrating these hypothetical relationships is shown in Fig 1.

The unique contribution of this study is largely twofold. On the one hand, this study attempts to understand the underlying mechanisms by which work–family conflict affects parent–child relationships by introducing the variable of parenting burnout. On the other hand, this study focuses on individual differences among people, i.e., the level of self-compassion, to verify whether this positive intraindividual resource is able to buffer the adverse effects of WFC.

## Results

### Descriptive statistics and correlation analysis

As shown in Table 1, work–family conflict was significantly negatively correlated with the parent–child relationship ($r = -0.41$, $p < 0.001$) and self-compassion ($r = -0.32$, $p < 0.001$), whereas it was significantly positively correlated with parenting burnout ($r = 0.37$, $p < 0.001$).

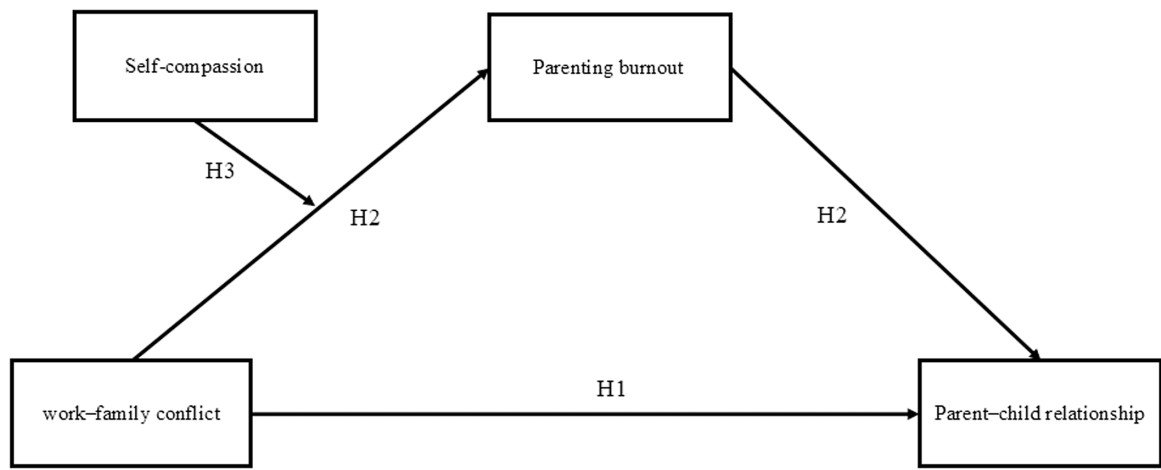

**Fig 1. Hypothesized research model.** Note. H1 = Hypothesis 1; H2 = Hypothesis 2; H3 = Hypothesis 3.

**Table 1. Descriptive statistics and correlation analysis for each variable.**

| Variable | M | SD | 1 | 2 | 3 | 4 | 5 |
|---|---|---|---|---|---|---|---|
| 1. Number of children | 1.59 | 0.56 | 1 | | | | |
| 2. Work-family conflict | 3.05 | 1.53 | −0.01 | 1 | | | |
| 3. Parent-child relationship | 3.77 | 0.59 | −0.11** | −0.41*** | 1 | | |
| 4. Parenting burnout | 2.05 | 1.15 | 0.07* | 0.37*** | −0.68*** | 1 | |
| 5. Self-compassion | 3.53 | 0.55 | −0.04 | −0.32*** | 0.56*** | −0.50*** | 1 |

Note. $M$ = mean, SD = standard deviation;

* $p < 0.05$,

** $p < 0.01$,

*** $p < 0.001$; all results are two-sided tests; the same notes apply to the following tables.

Parenting burnout was significantly negatively correlated with the parent-child relationship ($r = -0.68$, $p < 0.001$) and self-compassion ($r = -0.50$, $p < 0.001$). Self-compassion was significantly positively correlated with the parent-child relationship ($r = 0.56$, $p < 0.001$). Furthermore, the number of children demonstrated a significant negative correlation with the parent–child relationship ($r = -0.11$, $p = 0.002$) and a significant positive correlation with parenting burnout ($r = 0.07$, $p = 0.037$).

### Moderated mediation analysis

As presented in **Table 2**, work–family conflict had a significant direct effect on the parent–child relationship ($\beta = -0.16$, $p < 0.001$, 95% CI = [−0.18, −0.13]). After the incorporation of parenting burnout as a mediator and self-compassion as a moderator, work–family conflict was found to significantly positively predict parenting burnout ($\beta = 0.16$, $p < 0.001$, 95% CI = [0.12, 0.21]) and to significantly negatively impact the parent–child relationship ($\beta = -0.07$, $p < 0.001$, 95% CI = [−0.09, −0.05]). Furthermore, parenting burnout significantly negatively influenced the parent–child relationship ($\beta = -0.31$, $p < 0.001$, 95% CI = [−0.12, −0.02]). The bootstrap 95% confidence interval for the mediating effect of parenting burnout did not include 0, indicating that parenting burnout partially mediated the relationship between work–family conflict and the parent–child relationship.

**Table 2. Test of the moderated mediation model.**

| Regression equation | | Overall fit indicators | | | Regression coefficient significance | | | |
|---|---|---|---|---|---|---|---|---|
| Outcome variable | Predictor variable | R | R2 | F | β | BootLLCI | BootULCI | t |
| PC | NC | 0.42 | 0.18 | 87.65*** | −0.12 | −0.19 | −0.05 | −3.50*** |
| | PB | | | | −0.16 | −0.18 | −0.13 | −12.79*** |
| PC | NC | 0.70 | 0.49 | 264.40*** | −0.07 | −0.11 | 0.02 | −2.52** |
| | WFC | | | | −0.07 | −0.09 | −0.05 | −6.68*** |
| | PB | | | | −0.31 | −0.12 | −0.02 | −22.55*** |
| PB | NC | 0.58 | 0.33 | 104.00*** | −0.10 | −0.01 | 0.22 | 1.75 |
| | WFC | | | | 0.16 | 0.12 | 0.21 | 7.08*** |
| | SC | | | | −0.90 | −1.03 | −0.79 | −14.43*** |
| | WFC × SC | | | | −0.24 | −0.32 | −0.17 | −6.54*** |

Note: NC = the number of children; PC = the parent-child relationship; WFC = work–family conflict; PB = parenting burnout; SC = self-compassion.

**Table 3. Analysis of the mediating role of parenting burnout at different levels of self-compassion.**

| Mediating variable | Self-compassion | Indirect effect value | BootSE | BootLLCI | BootULCI |
|---|---|---|---|---|---|
| Parenting burnout | $M − 1SD$ | −0.09 | 0.01 | −0.12 | −0.07 |
| | M | −0.05 | 0.008 | −0.07 | −0.04 |
| | $M + 1SD$ | −0.008 | 0.008 | −0.02 | 0.007 |

Furthermore, the interaction of work–family conflict and self-compassion significantly and negatively predicted parenting burnout ($β = −0.24$, $p < 0.001$, 95% CI = [−0.32, −0.17]), thus suggesting that self-compassion plays a moderating role in the relationship between work–family conflict and parenting burnout.

To explore this moderating effect, a simple slope test and simple effects analysis were conducted using self-compassion values one standard deviation above and below the mean (see **Table 3** and **Fig 2**). When self-compassion was low (−1$SD$), work–family conflict significantly and positively predicted parenting burnout (simple slope = 0.30, $t = 10.30$, $p < 0.001$, 95% CI = [0.24, 0.35]). However, when self-compassion was high (+1$SD$), the positive prediction effect was no longer significant (simple slope = 0.03, $t = 0.79$, $p = 0.42$, 95% CI = [−0.04, 0.08]), as presented in Table 3. These results indicate that self-compassion mitigated the harmful effect of work–family conflict on parenting burnout.

# Discussion and implications

## Discussion

This study aimed to construct a moderated mediation model to investigate the destructive role of parenting burnout and the protective role of self-compassion in the relationship between WFC and the parent-child relationship. The results demonstrated that WFC affected the parent-child relationship both directly and indirectly and that parenting burnout served as a mediator in this relationship. Additionally, self-compassion was revealed to moderate the initial segment of the mediation pathway. These results confirm the previously proposed hypotheses and provide deeper insights into the mechanisms by which WFC influences the parent-child relationship; they also suggest potential intervention strategies aimed at improving the quality of the parent-child relationship.

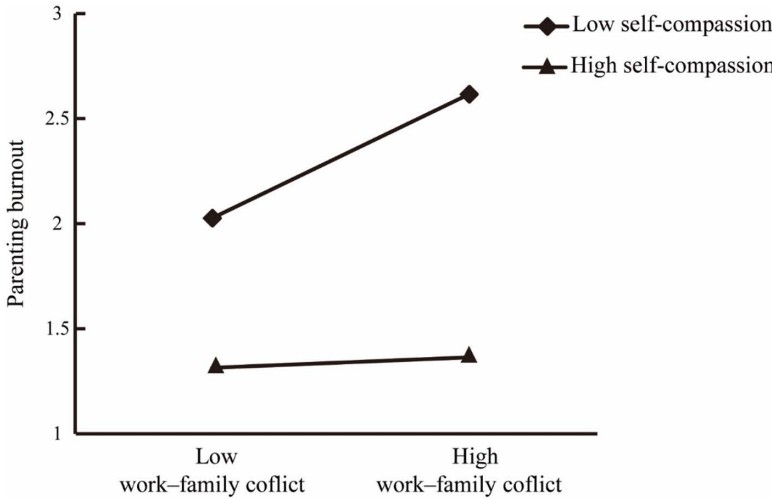

**Fig 2. Self-compassion moderates the effect of work–family conflict on parenting burnout.**

Consistent with previous studies, this study revealed that WFC is a significant risk factor for the parent-child relationship [8,33,34]. According to COR theory [19], individuals have limited resources, which they must allocate across their various roles to meet various needs. Therefore, an increase in the resources devoted to one role decreases the resources available to devote to other roles [35]. Accordingly, once parents allocate more resources to work with the goal of meeting the demands of their working role, they inevitably reduce the resources allocated to their family. Ilies reported that people facing severe work–family conflict tend to reduce their social interactions with family members [36]. Additionally, Roeters reported that time spent at work, which is a significant predictor of WFC [3], is negatively associated with the frequency of parent-child activities [33]. This finding implies that working parents must invest more resources in their work to meet rigid work demands while experiencing a decline in the quality of the parent-child relationship due to a reduction in family resources.

This study also revealed that WFC may increase the level of parenting burnout, thereby indirectly reducing the quality of the parent-child relationship. According to COR theory, individuals have limited resources to cope with stress, and the task of balancing multiple roles, such as those pertaining to work and family, often requires the allocation of significant resources to one role at the expense of the others [17,35]. This imbalance highlights the relationship between WFC and parenting burnout. Parenting burnout, in turn, can adversely affect the parent-child relationship in different family structures [14,15]. Emotional dysfunctions such as parenting burnout from the parenting subsystem can affect the parenting subsystem, as emphasized by family systems theory [37]. That is, negative emotions in the parental subsystem can spill over into the parent-child relationship, thereby exacerbating relationship challenges [38,39]. Working parents find it difficult to balance their work and family roles, which can lead to parenting burnout and other negative emotions, and if this day-to-day parenting burnout is not addressed, the parent-child relationship becomes extremely vulnerable to adverse effects.

Interestingly, our findings indicate that self-compassion plays a significant role in reducing the impact of work–family conflict on parenting burnout. Specifically, regardless of whether work–family conflict is high or low, parents with greater self-compassion consistently report significantly lower parenting burnout compared to those with lower self-compassion. This

finding suggests that self-compassion acts as a protective factor against the harmful effects of work–family conflict on parenting burnout. It may be that parents with greater self-compassion are more likely to manage work–family conflict effectively through practices such as self-kindness, maintaining a positive outlook, and recognizing shared humanity, particularly during challenging situations. These strategies help prevent excessive resource depletion [23,40], thereby lowering the risk of parenting burnout associated with work–family conflict. In contrast, parents with lower self-compassion struggle to manage work–family conflict proactively; consequently, these conflicts continually drain their resources, ultimately leading to greater parenting burnout. These findings highlight the potential for fostering self-compassion in working parents as a strategy to mitigate the risks of parenting burnout posed by work–family conflict. Previous researchers have explored numerous effective ways of increasing individuals' self-compassion [41,42], and we believe that this approach could be an effective way of improving the parent–child relationship.

## Theoretical implications

The results of this study carry significant theoretical implications. First, the study significantly extends the understanding of WFC by introducing self-compassion as a moderating variable. Traditional models often focus on external resources, such as social support, to alleviate the effects of WFC. However, by incorporating self-compassion—a personal, internal resource—the study highlights a new dimension in managing work-related stress and its influence on family dynamics. This study supports COR theory by suggesting that personal traits can play a crucial role in preserving psychological resources and mitigating stress. Moreover, this study provides further support for the spillover–crossover model, which suggests that stress originating in one domain (such as work) can spill over into another domain (such as family), ultimately impacting not just the individual but also those in their immediate environment. The findings offer empirical evidence that WFC contributes to parenting burnout, which subsequently harms the parent–child relationship. This finding highlights the interconnected nature of different life domains and emphasizes that individual well-being is shaped by the dynamic interplay of multiple factors across both work and family contexts. Third, by examining how parenting burnout mediates the relationship between WFC and parent–child interactions, this study applies family systems theory in a novel way. Disruptions in one part of the family system (e.g., the parent's work life) can lead to emotional and relational disruptions in another part (e.g., the parent–child relationship); this contributes to the theoretical understanding of how different subsystems within a family interact and influence each other, providing a more holistic view of family dynamics under stress.

## Practical implications

Beyond its theoretical contributions, this study offers several practical implications. The findings indicate that promoting self-compassion among working parents may serve as an effective approach to mitigating the adverse effects of WFC. Practitioners such as counselors and therapists can incorporate self-compassion training into their interventions. Techniques such as mindfulness-based stress reduction (MBSR) and self-compassion exercises could help individuals develop more compassionate inner dialog, reduce self-criticism, and better manage the demands of balancing work and family roles. Additionally, the study underscores the need for employers to acknowledge and address WFC, not only as an issue affecting employees' productivity but also as a factor influencing their overall well-being and family dynamics. Employers could consider implementing family-friendly policies, such as flexible work hours, telecommuting options, and onsite childcare services, to alleviate the stressors that

contribute to WFC. Moreover, providing resources and workshops on self-compassion and stress management could be beneficial in supporting employees' mental health. Furthermore, the findings underscore the need for support systems tailored specifically to working parents, especially mothers who may experience higher levels of WFC. Support groups, parenting resources, and community programs that focus on stress management and parenting strategies could be developed to assist parents in managing the dual demands of work and family more effectively. Finally, at a broader societal level, this study could inform advocacy efforts aimed at changing cultural norms and expectations around gender roles. By highlighting the detrimental effects of rigid gender expectations and the dual pressure on working parents to succeed both at work and at home, the findings could support initiatives promoting gender equality and more balanced domestic responsibilities. Limitations of the study and further research

## Limitations of the study and further research

Several limitations should be acknowledged in this study, along with suggestions for future research to address these issues. First, the cross-sectional design employed in this study restricts the capacity to infer causal relationships or capture the dynamic interplay among variables. As a result, definitive conclusions regarding the mechanisms by which WFC influences the parent–child relationship cannot be drawn. Future research could overcome this limitation by employing longitudinal or experimental designs, which would allow for a more in-depth exploration of these mechanisms over time. Second, the reliance on self-report questionnaires introduces the possibility of social desirability biases, potentially leading to unstable factors in the results. To mitigate these biases, future studies could incorporate additional methods, such as observational data or third-party reports, to strengthen the reliability of the findings. Furthermore, the reliance on convenience sampling instead of fully random sampling may restrict the generalizability of the results. Expanding the sample scope and employing random sampling techniques in future research could improve the the representativeness of the outcomes. Finally, since the data were provided predominantly by working parents, the study's perspective on how WFC affects the parent–child relationship may be somewhat narrow. The inclusion of data from children in future studies would provide a more holistic understanding of the dynamics within the parent–child system, thereby enhancing the representativeness and richness of the sample.

## Acknowledgments

The authors extend their gratitude to all participants for their willingness to take part in this study.

## Author contributions

**Conceptualization:** Zongping Chen.

**Data curation:** Zongping Chen.

**Formal analysis:** Zongping Chen.

**Funding acquisition:** Jing Liang.

**Methodology:** Zongping Chen.

**Supervision:** Jing Liang.

**Validation:** Zongping Chen.

**Writing – original draft:** Jing Liang, Zongping Chen.

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
