## [Decision Letter · Decision Letter 0]

6 Jan 2025

PONE-D-24-51556Parents’ work–family conflict and parent‒child relationship: The mediating role of parenting burnout and the moderating role of self-compassionPLOS ONE

Dear Dr. Liang,

Thank you for submitting your manuscript to PLOS ONE. After careful consideration, we feel that it has merit but does not fully meet PLOS ONE’s publication criteria as it currently stands. Therefore, we invite you to submit a revised version of the manuscript that addresses the points raised during the review process.

We look forward to receiving your revised manuscript.

Kind regards,

Fatma Refaat Ahmed, Ph.D.

Academic Editor

PLOS ONE

https://www.tandfonline.com/doi/full/10.1080/17439760.2014.936967

https://www.frontiersin.org/journals/psychology/articles/10.3389/fpsyg.2020.595987/full

In your revision ensure you cite all your sources (including your own works), and quote or rephrase any duplicated text outside the methods section. Further consideration is dependent on these concerns being addressed.

3. Thank you for stating the following financial disclosure:  [the 2024 Chongqing Humanities and Social Sciences Research Program (24SKSZ021) and the 2021 Chongqing Education Reform Program (213136)]. 

4. In the online submission form, you indicated that [The datasets analyzed during the current study are not publicly available but are available from the corresponding author on reasonable request.].

7. Please ensure that you refer to Figure 2 in your text as, if accepted, production will need this reference to link the reader to the figure.

Reviewers' comments:

Reviewer's Responses to Questions

**Comments to the Author**

1. Is the manuscript technically sound, and do the data support the conclusions?

Reviewer #1: Yes

Reviewer #2: Yes

Reviewer #3: Yes

2. Has the statistical analysis been performed appropriately and rigorously? 

Reviewer #1: Yes

Reviewer #2: Yes

Reviewer #3: Yes

3. Have the authors made all data underlying the findings in their manuscript fully available?

Reviewer #1: Yes

Reviewer #2: Yes

Reviewer #3: Yes

4. Is the manuscript presented in an intelligible fashion and written in standard English?

Reviewer #1: Yes

Reviewer #2: Yes

Reviewer #3: Yes

5. Review Comments to the Author

Reviewer #1: Dear Author's,

Thank you for your efforts on this interesting topic. I appreciate the work you have put into the paper. Below are my comments and suggestions for revisions:

1. Heading "1.1 The Current Study": I believe it would be more appropriate for this section to be titled "Aim of the Study" and placed underneath the "Materials and Methods" section. This would align better with the logical flow of the manuscript.

2. Methods Section: As per the journal guidelines, the heading title of the "Methods" section should be revised to "Materials and Methods."

3. Figures: I noticed that Figure 1 is missing from the results section. Kindly include it. Also, Figure 2 is mentioned, but there is no corresponding figure or related paragraph to describe it. Please ensure the figure is included and properly referenced in the text.

4. Formatting: Please adjust the formatting of the sentences to ensure that they are aligned neatly with the margins of the document for consistency and a professional presentation.

I look forward to seeing the revised version of your manuscript.

Best regards,

Reviewer #2: The manuscript is well written in standard English language throughout the documents.

All the components of the manuscript are included in the documents.

The background of the study and need of the study with the purpose has been clearly written.

Research methodology is well explained. Study variables and their measurement are explained.

Data analysis, results, discussion, theoretical and practical implications are discussed.

Need for Correction: (1) work‒family conflict significantly and positively predicts Parent‒child relationship; in this hypothesis, is there positive or negative prediction in parent- child relationship should be clear.

Reviewer #3: Comments

Sl. No Contents Comments

1 Abstract Main Objective is missing

2 Introduction No comments

3 Methodology Elaborate the methodology

Reliability score

4 Result No comments

5 Discussion Check for English and grammar

Over all check for English and grammar

6. PLOS authors have the option to publish the peer review history of their article (what does this mean? ). If published, this will include your full peer review and any attached files.

**Do you want your identity to be public for this peer review?** For information about this choice, including consent withdrawal, please see our Privacy Policy .

Reviewer #1: **Yes: ** Amany Anwar Saeed Alabdullah

Reviewer #2: **Yes: ** Menuka Bhandari

Reviewer #3: **Yes: ** Vinod Vishnu Bagilkar

---

## [Author Response · Author response to Decision Letter 1]

17 Jan 2025

For editor comments:

Dear Fatma Refaat Ahmed,

Thank you for your email and for the opportunity to revise our manuscript titled "Parents’ work–family conflict and parent‒child relationship: The mediating role of parenting burnout and the moderating role of self-compassion" (PONE-D-24-51556), which is currently under consideration for publication in PLOS ONE.

We appreciate the time and effort that you and the reviewers have dedicated to providing comprehensive feedback on our work. We have carefully considered all the comments and suggestions and have made corresponding revisions to the manuscript. Below, we provide point-by-point responses to the comments, alongside explanations of the changes we have implemented in the revised manuscript. We hope that these revisions have satisfactorily addressed the concerns.

1. Style Requirements: We have ensured that our manuscript adheres to the PLOS ONE style requirements, including file naming conventions. We have used the provided style templates to format our manuscript appropriately.

2. Overlapping Text: We have thoroughly reviewed the manuscript and have made changes to ensure that all sources are properly cited. Duplicated text outside the methods section has been either quoted or rephrased to avoid any issues of overlap.

3. Financial Disclosure: We have updated our financial disclosure to reflect the role of the funders in our study. Statement: "The funders provided financial support for the research, including data collection and manuscript preparation."

4. Data Availability: In accordance with PLOS ONE's data policy, we have uploaded the datasets used in this study to the figshare public database, which can be accessed at https://figshare.com, DOI: 10.6084/m9.figshare.28190315. This ensures that the data underlying our findings are freely available to other researchers.

5. ORCID iD: The corresponding author's ORCID iD has been provided and validated in the Editorial Manager.

6. Ethics Statement: We have included a complete ethics statement in the 'Materials and Methods' section of our manuscript. The statement includes the full name of the ethics committee that approved our study, as well as the details regarding informed consent.

7. Figure Reference: We have ensured that Figure 2 is correctly referenced in the text. This will facilitate the linking of the figure for readers if the manuscript is accepted for publication.

8. Reference List: We have reviewed the reference list to ensure its completeness and accuracy.

We would be honored if you would consider our manuscript for publication in PLOS ONE. Thank you for your time and consideration. Please do not hesitate to contact us if you require any further information or if you have any questions about our submission.

Sincerely, Jing Liang

For reviewers:

Dear reviewers,

Thank you for your email and for the opportunity to revise our manuscript titled "Parents’ work–family conflict and parent‒child relationship: The mediating role of parenting burnout and the moderating role of self-compassion" (PONE-D-24-51556), which is currently under consideration for publication in PLOS ONE.

We sincerely appreciate the time and effort that you and the reviewers have dedicated to providing comprehensive feedback on our work. We have carefully considered all the comments and suggestions and have made corresponding revisions to the manuscript. Below, we provide point-by-point responses to the comments, along with explanations of the changes implemented in the revised manuscript. We hope that these revisions adequately address the concerns raised. The main corrections to the paper and our responses to the reviewers’ comments are as follows.

Point-by-point response to reviewer comments:

To Reviewer #1:

Reviewer comment: 1. Heading "1.1 The Current Study": I believe it would be more appropriate for this section to be titled "Aim of the Study" and placed underneath the "Materials and Methods" section. This would align better with the logical flow of the manuscript.

Response: Thank you very much for your insightful comment. We fully agree with your suggestion regarding the title and placement of this section. To align better with the logical flow of the manuscript, we have made the adjustment that you recommended. The section previously titled "1.1 The Current Study" has been renamed "Aim of the Study" and moved to the "Materials and Methods" section. We believe that this change will enhance the clarity and coherence of our manuscript. Thank you again for your valuable feedback.

Reviewer comment: 2. Methods Section: As per the journal guidelines, the heading title of the "Methods" section should be revised to "Materials and Methods."

Response: Thank you for bringing this to our attention. We apologize for this oversight and have revised the heading title of the section in accordance with the journal guidelines. The "Methods" section has been renamed to "Materials and Methods" as required. We appreciate your guidance in ensuring that our manuscript adheres to the journal's formatting standards.

Reviewer comment: 4. Formatting: Please adjust the formatting of the sentences to ensure that they are aligned neatly with the margins of the document for consistency and a professional presentation.

Response: Thank you for your suggestion regarding the formatting of our manuscript. We understand the importance of a neat and consistent presentation. To address this, we have carefully adjusted the formatting of all sentences, ensuring that they are now aligned neatly with the document margins. This adjustment has been made throughout the entire manuscript to maintain a professional appearance and enhance readability. We appreciate your attention to detail and believe that these changes will contribute to a more polished final document.

To Reviewer #2:

Reviewer comment: Need for Correction: (1) work‒family conflict significantly and positively predicts Parent‒child relationship; in this hypothesis, is there positive or negative prediction in parent- child relationship should be clear.

Response: Thank you for pointing out the need for clarification in our hypothesis regarding the relationship between work‒family conflict and the parent‒child relationship. Our original hypothesis was that "work‒family conflict significantly and positively predicts the parent‒child relationship." However, this phrasing may have been misleading, as it could be interpreted that a higher level of work‒family conflict leads to a better parent‒child relationship, which is not our intended meaning.

What we actually mean is that work‒family conflict has a significant effect on the parent‒child relationship. Specifically, as the level of work‒family conflict increases, the quality of the parent‒child relationship tends to decrease. In other words, there is a negative correlation between work‒family conflict and the parent‒child relationship. Greater work‒family conflict is associated with more problems or challenges in the parent‒child relationship rather than improvement.

To correct and clarify this, we have rephrased the hypothesis as follows:

"Work‒family conflict significantly and negatively predicts the parent‒child relationship."

This revised hypothesis accurately reflects our intended meaning that increased work‒family conflict is associated with a decline in the positive aspects of the parent‒child relationship, such as reduced emotional closeness, more conflict, or less effective communication between parents and children.

We apologize for any confusion caused by the original phrasing and appreciate your feedback in helping us improve the clarity and accuracy of our research.

To Reviewer #3:

For abstract:

Reviewer comment: 1 Abstract Main Objective is missing

Response: Thank you for pointing out that the main objective of the study was missing from the abstract. We apologize for this oversight. The main objective of our study is to explore the impact of work‒family conflict on the parent‒child relationship, with a focus on the mediating role of parenting burnout and the moderating role of self-compassion. We have now included this information in the abstract to provide a clear and concise overview of the purpose of our research.

For Methodology:

Reviewer comment: Elaborate the methodology

Response: Thank you for your suggestion to elaborate on the methodology. We have supplemented the methodology section with additional details to enhance the transparency and rigor of our study.

Reviewer comment: Reliability score

Response: Thank you for your comment. The reliability scores for each of the survey questionnaires are reported in the revised manuscript. Specifically, they can be found on lines 113 (work‒family conflict), 127 (parent‒child relationship), 135 (parental burnout), and 146 (self-compassion).

For Discussion:

Reviewer comment: Check for English and grammar

Response: Thank you for your valuable feedback and for pointing out the potential language issues in our manuscript. In response to your comment, we have meticulously checked the discussion section for English and grammar to increase its readability and ensure that it meets the standards of academic writing.

---

## [Decision Letter · Decision Letter 1]

6 Feb 2025

Parents’ work–family conflict and parent‒child relationship: The mediating role of parenting burnout and the moderating role of self-compassion

PONE-D-24-51556R1

Dear Dr. Liang,

We’re pleased to inform you that your manuscript has been judged scientifically suitable for publication and will be formally accepted for publication once it meets all outstanding technical requirements.

Kind regards,

Fatma Refaat Ahmed, Ph.D.

Academic Editor

PLOS ONE

Additional Editor Comments (optional):

Reviewers' comments:

Reviewer's Responses to Questions

**Comments to the Author**

1. If the authors have adequately addressed your comments raised in a previous round of review and you feel that this manuscript is now acceptable for publication, you may indicate that here to bypass the “Comments to the Author” section, enter your conflict of interest statement in the “Confidential to Editor” section, and submit your "Accept" recommendation.

Reviewer #1: All comments have been addressed

Reviewer #3: All comments have been addressed

2. Is the manuscript technically sound, and do the data support the conclusions?

Reviewer #1: Yes

Reviewer #3: Yes

3. Has the statistical analysis been performed appropriately and rigorously? 

Reviewer #1: Yes

Reviewer #3: Yes

4. Have the authors made all data underlying the findings in their manuscript fully available?

Reviewer #1: Yes

Reviewer #3: Yes

5. Is the manuscript presented in an intelligible fashion and written in standard English?

Reviewer #1: Yes

Reviewer #3: Yes

6. Review Comments to the Author

Reviewer #1: (No Response)

Reviewer #3: No comments for the above study .....................................................................................................................

7. PLOS authors have the option to publish the peer review history of their article (what does this mean? ). If published, this will include your full peer review and any attached files.

**Do you want your identity to be public for this peer review?** For information about this choice, including consent withdrawal, please see our Privacy Policy .

Reviewer #1: **Yes: ** Amany Anwar Saeed Alabdullah

Reviewer #3: **Yes: ** Vinod V Bagilkar

---

## [Editor Report · Acceptance letter]

PONE-D-24-51556R1

PLOS ONE

Dear Dr. Liang,

I'm pleased to inform you that your manuscript has been deemed suitable for publication in PLOS ONE. Congratulations! Your manuscript is now being handed over to our production team.

Kind regards,

on behalf of

Dr. Fatma Refaat Ahmed

Academic Editor

PLOS ONE